# Prevalence and Risk Factors of Abdominal Aortic Aneurysms Detected with Ultrasound in Korea and Belgium

**DOI:** 10.3390/jcm12020484

**Published:** 2023-01-06

**Authors:** Hyangkyoung Kim, Sungsin Cho, Natzi Sakalihasan, Rebecka Hultgren, Jin Hyun Joh

**Affiliations:** 1Department of Surgery, Ewha Womans University College of Medicine, Ewha Womans University Medical Center, Seoul 07985, Republic of Korea; 2Department of Surgery, Kyung Hee University Hospital at Gangdong, Kyung Hee University School of Medicine, Seoul 05278, Republic of Korea; 3Department of Cardiovascular and Thoracic Surgery, Centre Hospitalier Universitaire Liège, University of Liège, 4000 Liège, Belgium; 4Department of Molecular Medicine and Surgery, Karolinska Institutet, 17177 Solna, Sweden; 5Department of Vascular Surgery, Karolinska University Hospital, 17164 Stockholm, Sweden

**Keywords:** abdominal aortic aneurysm, prevalence, risk factor, diameter, Asia, iliac aneurysm, population-based screening

## Abstract

The objective was to investigate the prevalence of abdominal aortic aneurysms (AAAs) and the diameters of the aorta and common iliac arteries (CIAs) in a Korean cohort and secondly to analyze the differences in aortic diameter by comparison with a European cohort. The Korean cohort included participants ≥ 50 years who consented to AAA screening and data were analysed retrospectively. Aortic and common iliac diameters were measured using the outer-to-outer diameter method and prevalence rates were calculated. Common risk factors such as smoking, body mass index, pulmonary disease, hypertension, diabetes, hyperlipidaemia, ischaemic heart disease, and cerebrovascular disease were reported in association with AAA occurrence and AAA development. The aortic diameters were then compared with those in a Belgian cohort of 2487 participants identified in the Liège AAA Screening Program. An aortic size index (ASI) was also calculated to account for the potential size differences in the Belgian and Korean populations. A total of 3124 Korean participants were examined using ultrasound. The prevalence of AAAs in this cohort was 0.7%. The combined prevalence of subaneurysmal dilatation and AAA was 1.5%. The prevalence in male smokers older than 65 years was 2.7% (19/715). The mean infrarenal aortic diameter was 17.3 ± 3.1 mm in men and 15.7 ± 2.7 mm in women; the corresponding values in Belgian participants were 19.4 ± 3.0 mm in men and 17.9 ± 2.4 mm in women. The median aortic size index was 0.99 (interquartile range 0.88–1.12). The mean infrarenal aortic diameter was significantly smaller in the Korean cohort than in the Belgian cohort. Considering the observed prevalence of AAAs in different age groups, the age groups which would contribute to most cases was male persons above 66 years in both cohorts.

## 1. Introduction

Abdominal aortic aneurysm (AAA) is defined as a permanent localized dilatation of the abdominal aortic artery to ≥3.0 cm in diameter or at least a 50% increase in diameter compared to the normal diameter [1,2]. AAA screening programs that have been established in several countries have demonstrated effectiveness in reducing mortality by timely diagnosis and reducing health-related costs, and one-time AAA screening in men has resulted in significant reductions in the rates of AAA-related mortality and AAA rupture [3,4]. Due to the absence of the AAA screening program and also the lack of national easy-accessible registries, the rupture rate in Korea is not yet known.

For a successful surveillance program, it is essential to determine a suitable target population based on accurate data on prevalence and risk factors. The prevalence of AAAs varied from 1.34% to 10% in Western countries when using ultrasound as screening [5,6]. Previous smaller studies in high-risk Koreans (male > 65 years, smoker, family history of AAA) revealed a AAA prevalence of 3.2–4.5%, and in the general population 2% in men and 0.4% in women [7,8]. In nationwide screening programs, the prevalence was 1.3–1.5%, with diagnostic cutoff values of 30 mm [5,9]. It is reported that the aortic diameter is correlated with body size [10]. The aortic size index (ASI) has been proposed to predict rupture risk based on the aortic diameter adjusted to the body surface area [11]. If we extrapolate this to Asian populations, an ASI ratio of 1.5 in a male would correspond to an aortic diameter of 30 mm, and in females 26.3 mm. It should be noted that diameter-correlated rupture risks are based on autopsy studies performed on predominately Caucasian males [12]. This has been particularly emphasized for women with smaller body sizes, but it is possible that the same should be considered for Asian populations.

During the last two decades, while the prevalence and incidence of AAAs have decreased in developed countries, the prevalence is increasing in some countries in Latin America and the Asia Pacific region [13]. It should be noted that overall, there is a lack of robust knowledge about the actual prevalence of AAAs globally, especially in asymptomatic Asians, as reports on the prevalence of AAAs detected using ultrasound screening programs are only available for a few countries [14]. South Korea is the fastest-aging country in the world, having become an aged society in 2017, with over 14% of its population aged 65 years or older [15]. Therefore, the aim of this study was to explore the diameters of the aorta and iliac arteries and the prevalence of AAAs in the Korean cohort. The secondary aim was to investigate risk factors associated with AAA development in this cohort. The third aim was to elucidate differences in aortic diameter through comparison with a Belgian cohort.

## 2. Materials and Methods

### 2.1. Study Cohorts

The Korean study cohort was recruited through voluntary participation, by visiting the Welfare Community Center in seven different cities in South Korea, namely, Gangdong-gu of Seoul, Hanam, Uiwang, Guri, Namyangju, Gosung, and Ulsan between 2008 and 2019. Recruitment was achieved with an official document of authority without any advertisement or reward. Subjects ≥ 50 years of age who consented to AAA screening were included in the study. The ultrasound examination was performed free of charge by a single team, and participation was achieved by promoting the program through flyers or posters by the administrative officials of each city to those living near the Welfare Community Center. Subjects who voluntarily consented to undergo an AAA ultrasound examination were included in the study. Established occlusive or aneurysmal diseases in the aorta were excluded.

Participants in the Belgian cohort were recruited through two separate screening programs established in Liège, Belgium. There were 796 women and 1691 men who participated, all of whom were Caucasian. Participants’ characteristics have been published in previous papers [16,17]. This study complied with the principles of the Declaration of Helsinki.

### 2.2. Definitions

The diagnostic cutoff values for indicating aneurysmal disease were as follows: 30 mm in men and 28 mm in women for the aorta and 18 mm in men and 15 mm in women for the common iliac artery (CIA) (calculated based on reporting standards [18]. Subaneurysmal aortic dilatation was defined as a maximum aortic diameter of 25–29 mm in men and 25–27 mm in women [2]. The patients’ medical and family history was obtained using a detailed questionnaire that included questions about hypertension, diabetes, hyperlipidemia, history of cardiovascular disease, pulmonary disease, cerebrovascular disease, renal function impairment, and prior surgery. Height and weight were measured. Body surface area (BSA) was calculated using the Dubois and Dubois formula [19]. The aortic size index (ASI) was calculated by dividing the aneurysm diameter (in cm) by BSA [11]. Hypertension and hyperlipidemia were defined as the intake of antihypertensive and lipid-lowering medications, respectively. Individuals taking anti-diabetic medication or regularly visiting a doctor because of elevated fasting plasma glucose levels were considered to have diabetes. Cardiovascular diseases (CVD) included arrhythmia, coronary artery disease, myocardial infarction, angina, and a history of coronary angioplasty or stenting. Cerebrovascular diseases included transient ischemic attack, reversible ischemic neurologic deficit, and cerebrovascular accidents. Details of health-related behaviors, including smoking, alcohol consumption, and regular exercise, were also recorded. Pulmonary diseases included chronic obstructive pulmonary disease, and pulmonary tuberculosis. Chronic kidney disease (CKD) was defined as any condition that permanently diminished renal function and/or required regular dialysis.

### 2.3. Ultrasound Examination Procedure

Ultrasound examination in the Korean cohort was performed using the same strategy as that described in our previous study [7]. In brief, the aortic diameter was measured at four sites: suprarenal, for the diameter above the higher renal artery; pararenal, for the diameter at the renal artery level; infrarenal, for the maximal diameter at the site between the renal artery level and the bifurcation; and CIA, for the maximal diameter of each CIA. Representative diameters at the five levels were calculated as the mean values of the two anteroposterior measurements at each level. The maximal diameters of the suprarenal, pararenal, infrarenal, and both CIAs were measured according to the outer-to-outer principle, which has high reproducibility [20]. Duplex scanning was performed by experienced vascular surgeons. Three types of ultrasound machines were used: Zonare (Zonare Medical Systems, Mountain View, CA, USA), VIVID e (GE, Chicago, IL, USA), and HD7 (Philips, Amsterdam, The Netherlands). A 2.5–5 MHz convex ultrasound probe was used for the examination. Duplex scanning was performed after the patient had fasted for at least 8 h. Duplex scanning was performed from the infradiaphragmatic level to the bilateral iliac arteries. The prevalence of aneurysmal disease and clinical characteristics were analyzed. The representative diameter values were obtained after excluding aneurysms. The measured diameters were compared between men and women, and by age. The infrarenal aortic diameter was then compared between our cohort and the Belgian cohort (n = 2047), stratified by sex and age. In the Belgian cohort, each participant underwent an ultrasound (US) of the abdominal aorta using a portable US machine (GE Logiq e Ultrasound, Transducer: curved array 2–5 MHz). The US examinations were performed by the same investigator (with 3 years of experience in aortic US examination) and validated by an experienced physician.

### 2.4. Statistics

The statistical analysis was performed using the SPSS version 25.0 software (Armonk, NY, USA). Data with normal distribution were summarized as mean value ± standard deviation, and non-normally distributed data were summarized as median and interquartile ranges (IQRs). Fisher’s exact test was used to analyze the univariate associations of categorical variables with respect to AAAs. From the initial results of the univariate analysis, variables with *p* < 0.20 were incorporated into the multivariate model (binary logistic regression for the dichotomized endpoints). Mean diameters in different age groups were analyzed using analysis of variance, and Tukey’s b test was used for post-hoc analysis. Linear and multilinear analyses with stepwise selection were performed to analyze the association between the variables and arterial diameters. To compare the Korean and Belgian cohorts, a one-sample t-test was adopted. Statistical significance was set at *p* < 0.05.

## 3. Results

### 3.1. Characteristics of Study Cohorts

The baseline demographic data of the Korean and Belgian cohorts are summarized in Table 1. We included 3124 Korean participants, 43.4% men. There were 13 participants with missing data. There were more women in the over 65 years age group (46.1% vs. 53.9%, *p* < 0.001). There was a significant difference in the median age between the two cohorts: Korean, 69 years, and Belgian, 72.4 years (*p* < 0.001). There was a significant difference in the body mass index (BMI), BSA and in the prevalence of underlying conditions including hypertension, DM, hyperlipidemia, CVD, pulmonary disease, cerebrovascular disease, CKD, and smoking.

### 3.2. Diameters

After excluding aneurysms, the mean aortoiliac diameters in the Korean cohort were as follows: suprarenal, 19.7 ± 2.8 mm; pararenal, 18.4 ± 3.0 mm; infrarenal, 17.3 ± 3.1 mm; and CIA, 10.4 ± 2.2 mm in men and suprarenal, 18.7 ± 2.9 mm; pararenal, 16.8 ± 2.6 mm; infrarenal, 15.7 ± 2.7 mm; and CIA, 9.8 ± 2.0 mm in women. The mean infrarenal aortic diameter and CIA diameter in participants without AAAs were analyzed according to age group (Figure 1 for the mean infrarenal aortic diameter and Figure 2a for the CIA diameter).

In the men, no significant difference was found in the aortic diameter between different age groups, whereas, there was a significant difference in the CIA diameter (*p* < 0.001). Post-hoc analysis revealed a significant difference in CIA diameter between participants over 65 years of age and participants under 65 years of age. In the women, significant differences were found in the diameters of the aorta and CIA among different age groups (aorta *p* = 0.001, CIA, *p* < 0.001). The aortic diameter was smaller in the under-50 years age group than in the over-50 years age group. The CIA diameter was smaller in the older age group. There was a negative linear correlation between age and mean CIA diameter (*p* < 0.001) and a negative correlation between age and body surface area, (*p* < 0.001) which had the strongest positive correlation with an arterial diameter (Figure 2b,c). The median ASI was 0.99 (IQR 0.88–1.12). There was no significant difference in the aortic index between men and women; 0.98 (IQR 0.86–1.10) in men and 1.00 (IQR 0.88–1.12) in women (*p* = 0.058).

### 3.3. Prevalence of AAAs and Risk Factors for AAAs

AAAs were detected in 22 individuals (0.7%) and CIA aneurysms were detected in 54 (1.7%) individuals in the Korean cohort (Table 2). The prevalence of AAAs was significantly higher in men (1.5%) than in women (0.1%) (*p* < 0.001). The median age was significantly higher in participants with AAAs (70 (IQR 63–75). vs. 74.5 (IQR 70.75–80.25). years, *p* < 0.001) than in those without AAAs. Participants aged > 65 years had a significantly higher prevalence of AAAs (20/2087 (1.0%), *p* = 0.02), and there were 19 men aged > 65 years with a smoking history (19/22 (86.4%)). Subaneurysmal dilatation was detected in 16 men (1.2%) and in 10 women (0.6%). The combined prevalence of subaneurysmal dilatation and AAA was 1.5%. ASI was greater than or equal to 1.5 in a total of 55 (1.8%) patients, of which 32 were men and 23 were women.

The underlying conditions and age distribution of participants with AAAs in the Korean cohort and the Belgian cohort are depicted in Figure 3. Of the 22 AAAs in the Korean cohort, there were nine with AAAs of 40–50 mm, one with AAA of 50–55 mm, and three with AAAs larger than 55 mm. The age of the participants with AAAs ranged from 64 to 85 years. The number of individuals in each age group was as follows; two in 60–65 years old, three in 66–70 years old, 12 in 71–80 years old, and five in over 81 years old. There were 51 AAAs in the Belgian cohort, and there were three with AAAs of 40–50 mm, two with AAAs of 50–55 mm, and two with AAAs larger than 55 mm. In the Belgian cohort, the age range of the participants with AAAs was from 66 to 85 years. The number of individuals in each age group was as follows; nine in 66–70 years old, 28 in 71–80 years old, and 14 in over 81 years old. All participants completed the questionnaires and underwent the examination. The underlying conditions of the participants with AAAs are shown in Table 3. The prevalence of AAAs was 1.5% in men and 0.06% in women, with a significant difference between sexes (*p* < 0.001). The prevalence of AAAs in men aged > 65 years with a smoking history was 2.7% (19/715). Patients with AAA had a significantly higher rate of smoking history (20/1089 [1.84%], *p* < 0.001).

Women had a significantly reduced risk for AAAs in the unadjusted model (Table 3). Older age (odds ratio (OR) 1.096 (1.037–1.159), *p* = 0.001), CKD (OR 6.135 (1.694–22.227), *p* = 0.006), and smoking history (OR 5.874 (1.054–32.733), *p* =0.043) were associated with an increased risk of AAAs.

In the linear analysis, female sex, age, BMI, BSA, DM, hyperlipidemia, CVD, smoking history, and alcohol consumption were significantly associated with an aortic diameter (*p* < 0.05) (Table 4). In the multilinear analysis model, all variables except age, diabetes mellitus, hyperlipidemia, and smoking history significantly predicted aortic diameter (*p* < 0.0001). All significant variables except the female sex were associated with increased aortic diameter.

### 3.4. Comparisons with the Belgian Population

The Belgian cohort consisted of 1691 men and 796 women. The mean suprarenal aortic diameter in the Belgian cohort was 19.7 ± 2.7 mm in men and 18.0 ± 2.9 mm in women. In women, the suprarenal aortic diameter was significantly larger in the Korean cohort than in the Belgian cohort (*p* < 0.001), while no significant difference was observed in men (*p* = 0.861). The mean infrarenal aortic diameter in the Belgian cohort was 19.4 ± 3.0 mm in men and 17.9 ± 2.4 mm in women. When comparing the Korean and Belgian cohorts, the infrarenal aortic diameter was smaller in the Korean cohort in all age groups, regardless of sex (all *p* < 0.001, Figure 4).

Upon comparison according to sex, the aortic diameter in male participants in the Korean cohort was smaller (odds ratio 1.768, 95% confidence interval (CI) −2.007, −1.530, *p* < 0.001), and the aortic diameter in female participants was also smaller in the Korean cohort by 1.51 mm (95% CI −1.64, −1.38, *p* < 0.001). In participants aged ≤ 65 years, the difference was −1.53 mm (95% CI −1.84, −1.23, *p* < 0.001) in men and −2.27 mm (95% CI −2.46, −2.08, *p* < 0.001) in women. In participants aged > 65 years, the corresponding differences were −1.74 mm (95% CI −2.06, −1.43, *p* < 0.001) and −1.42 mm (95% CI −1.59, −1.25, *p* < 0.001), respectively. The BMI was also lower in Koreans in both men and women, and the same results were obtained in each age group (*p* < 0.001). In the Belgian cohort, AAAs were detected in 51 individuals (3.2%), and the prevalence of AAAs in the men and women was 4.2% (45/1068) and 1.2% (6/205), respectively (Table 2). Body surface area was smaller in the Korean cohort in both men and women. (*p* < 0.001, respectively, Figure 5).

## 4. Discussion

In this ultrasound-based examination of an invited Korean cohort, the prevalence of AAA was low, but there was a considerable rise in the prevalence of AAAs if the ASI threshold for AAA was applied. In the risk group analysis, the prevalence was much higher in men, age > 65, and smokers, similar to that shown in previous reports. These findings also indicate that a different baseline threshold for AAA definitions should be used for the Asian cohort examined, considering the much smaller aortic and iliac diameters. According to the literature, the prevalence of AAAs > 30 mm varied from 1.5% to 10% in patients aged > 50 years in Western countries in an ultrasound screening and autopsy series [2,4,5]. There is a paucity of solid prevalence data in Asian populations, and only selected attempts to screen cohorts have been performed before with variations in reported prevalence rates [21]. The prevalence was relatively low in our patients, even when combining subaneurysmal aortic dilatation. Although a convincing explanation for this finding is not apparent in our study, racial differences, volunteer bias in the recruitment process, and selection bias due to the small sample size may be the reasons for this difference. In particular, the prevalence of diabetes in this cohort was higher than the known prevalence rate in the general Korean population at 14.4% as well as that in the Belgian group, which would lower the prevalence rate [2,22,23]. Although a statistically significant difference in AAA prevalence was not found between patients with and without diabetes in this study, a protective role of diabetes for AAAs could be considered as one of the causes [24]. The prevalence of AAAs in our cohort was 0.7%, this increased to 2.7% when we included a high-risk group of men over the age of 65 who smoked. However, the 0.1% of clinically relevant AAAs (>55 m) was 0.4% in the high-risk population of >65-year-old smokers. Previous studies on small populations showed that the prevalence of AAAs was 3.2–4.5% in the high-risk group, which includes those with a family history of AAA and men 65–75 years who have smoked at least 100 cigarettes throughout their life [7,8]. Among male smokers, the prevalence of AAAs was 2.7% in our study, that is, it was almost four times higher than that in nonsmokers, and a similar order of magnitude was noted in previous studies. The low prevalence of AAA in our study may have limitations in providing clear evidence for the risk factors of AAA. However, it seems noteworthy that men with a smoking history had a relatively high prevalence of AAAs, especially considering the fact that the smoking rate in Korea is one of the highest among those in the Organization for Economic Cooperation and Development countries [25]. The US Preventive Task Force suggested routine screening for men aged 65–75 years with a smoking history based on the evidence that AAA screening in men aged 65–75 years does not appear to be associated with significant physical or psychological harm and that it reduces AAA-related mortality [26]. The target population for AAA screening is somewhat different between different societies, but all populations have similar epidemiological characteristics [23,27]. It is necessary to determine the screening strategy based on a nationwide prevalence study.

When we compared the Korean cohort, after excluding AAA patients, with the Belgian cohort, the mean infrarenal aortic diameter, and BMI according to age and sex were significantly smaller and lower, respectively, in the Korean cohort. This difference may have been observed because participants in the Belgian group were older, but the same result was also found when comparing the same age groups. As a significant correlation was found between BMI and aortic diameter, the difference between the Korean and Belgian cohorts was not surprising. Depending on the sex, especially in women, the aortic diameter indexed to body size was suggested to be particularly important in terms of predicting rupture risk of aneurysm, and the suggested cut-off to define AAA would be an ASI of 1.5 corresponding to 30 mm AAA in men [11]. An interracial difference in the reference aortic diameter is controversial [28,29,30]. Previous studies have reported a significantly smaller diameter in Asians, which is in line with our findings. However, further investigation into the clinical significance of this finding is needed to determine whether different diagnostic criteria should be applied for different races. The prevalence of AAAs was different between the 30 mm and ASI ≥ 1.5 threshold in both men and women (1.5% vs. 2.4% in men and 0.1% vs. 1.3% in women). Therefore, if a smaller criterion is applied to Asians with smaller aortic diameters, the prevalence of AAA is expected to be higher than the results of this study. Future studies to elucidate the actual risk of rupture in relation to the aortic diameter in the Asian population, as well as the threshold that signifies an indication for elective repair of AAAs in these subsets, need to be undertaken. Interestingly, the infrarenal aortic diameters in the age groups of 51–55 were similar in the Korean and Belgian cohorts, as compared to the older age groups which showed larger differences (Figure 4a). Age seemed to affect the aortic diameter of the Belgian cohort more than the Korean cohort.

This study had some limitations. First, this study used a small sample size, including individuals from only a few provinces in Korea. Another potential source of bias in our study was selection bias. Especially, the proportion of participants with diabetes was higher than the prevalence in the general population. We recruited people from rural areas who were willing to participate in this study. It is possible that more people with greater concern for their health and fewer people undergoing regular hospital check-ups for underlying diseases might have been included in this study, which could have had an impact on the actual detection rate of AAAs.

## 5. Conclusions

The prevalence of AAAs was 0.7% and that of CIA aneurysms was 1.7%. The prevalence of AAAs in male smokers aged > 65 years was 2.7%. Our data support that the risk factor profile for the development of diseases in our cohort is similar to that previously reported for AAA, male sex, age, and smoking. The mean infrarenal aortic diameter was significantly smaller in the Korean cohort than that in the Belgian cohort. Considering the observed prevalence of AAAs in the high-risk group, the age range with the most yield for ultrasound screening was the over 66 years in both the Korean and Belgian cohorts.

## Figures and Tables

**Figure 1 jcm-12-00484-f001:**
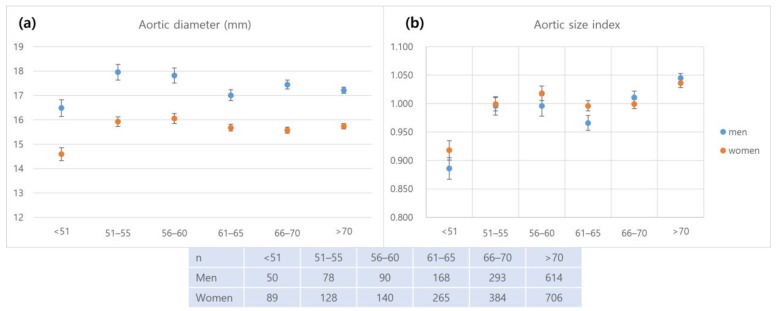
Distribution of the diameters of the aorta and aortic size index according to the age group in the Korean cohort: (**a**). Aortic diameter; (**b**) Aortic size index.

**Figure 2 jcm-12-00484-f002:**
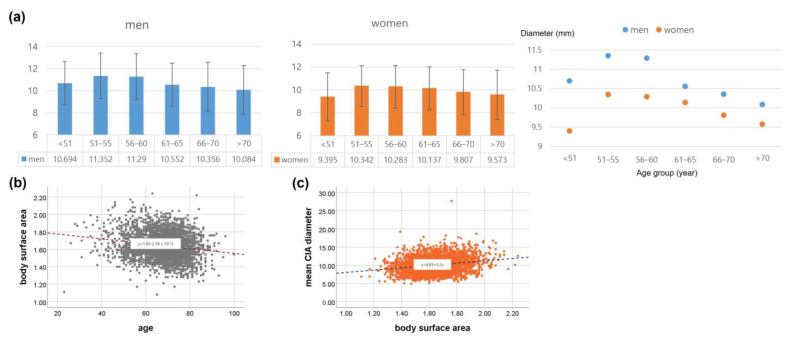
Distribution of the diameters of the common iliac arteries (CIAs) in the Korean cohort: (**a**) diameters of the CIAs of men and women; (**b**) Relationship between age and mean CIA diameter; (**c**) Relationship between body surface area and mean CIA diameter.

**Figure 3 jcm-12-00484-f003:**
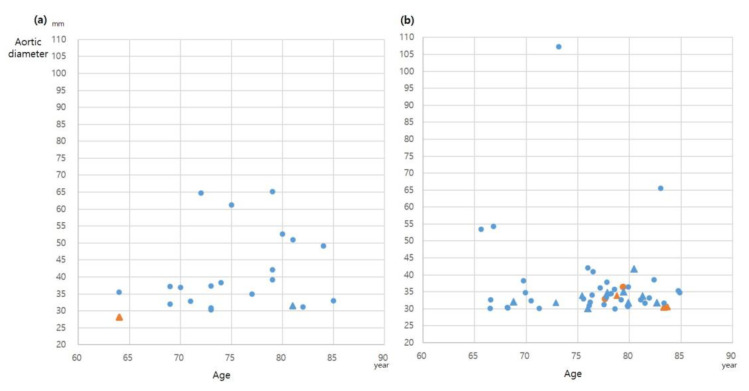
Distribution of participant age and diameter of abdominal aortic aneurysms: (**a**), Korean (n = 22); (**b**), Belgian (n = 51). blue, men; orange, women; circle, smoker; triangle, nonsmoker.

**Figure 4 jcm-12-00484-f004:**
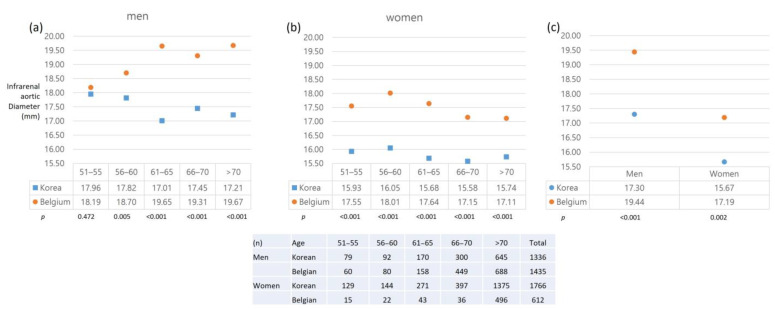
Comparison of the aortic diameter between the Korean and Belgian cohorts: (**a**), Infrarenal aortic diameter in men according to the age group; (**b**), Infrarenal aortic diameter in women according to the age group; (**c**) mean aortic diameter in men and women.

**Figure 5 jcm-12-00484-f005:**
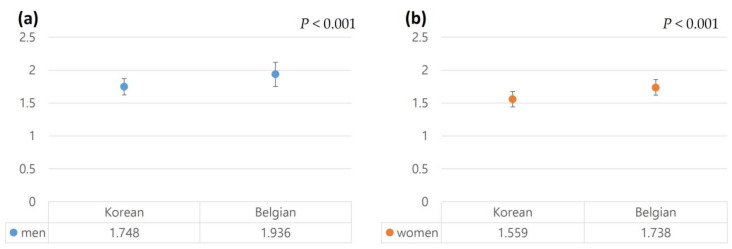
Body surface area of Korean and Belgian cohort: (**a**), Men; (**b**), Women. blue, men; orange, women.

**Table 1 jcm-12-00484-t001:** Demographic data of the Korean and Belgian cohorts.

		Korean	Belgium	*p*
N		3124	2487	
Age		69 (61–75)	72.4 (66.3–78.2)	<0.001
BMI		24.1 (23.3–26.0)	25.8 (23.5–28.4)	<0.001
BSA		1.63 (1.53–1.75)	1.87 (1.15–2.94)	<0.001
Hypertension		617 (19.8)	1516 (61.0)	<0.001
DM		1518 (48.6)	522 (21.0)	<0.001
Hyperlipidemia		1559 (49.9)	1469 (59.1)	<0.001
CVD		806 (25.8)	785 (31.5)	<0.001
Pulmonary disease		2 (0.1)	204 (8.2)	<0.001
Cerebrovascular disease		290 (9.3)	175 (7.0)	0.002
CKD		70 (2.2)	159 (6.4)	<0.001
Smoking	Current smoker	364 (14.6)	283 (20.9)	<0.001
	Ex-smoker	1028 (41.3)	196 (14.4)	

BMI, body mass index; BSA, body surface area; DM, diabetes mellitus; CVD, cardiovascular disease; CKD, chronic kidney disease.

**Table 2 jcm-12-00484-t002:** Prevalence of abdominal aortic and iliac artery aneurysms in the Korean cohort and abdominal aortic aneurysms in the Belgian cohort.

Prevalence of abdominal aortic aneurysms in men			
		Korean, n = 1357	Belgian, n = 1068
Abdominal aortic aneurysm		21 (1.5)	45 (4.2)
Aortic size index ≥ 1.5		32 (2.4)	39 (3.7)
Age	≤65 years	1 (0.1)	1 (0.1)
	>65 years	20 (1.5)	44 (4.1)
Diabetes		12 (0.9)	9 (0.8)
Prevalence of abdominal aortic aneurysms in women			
		Korean, n = 1767	Belgian, n = 502
Abdominal aortic aneurysm		1 (0.1)	6 (1.2)
Aortic size index ≥ 1.5		23 (1.2)	6 (1.2)
Age	≤65 years	1 (0.1)	0
	>65 years	0	6 (1.2)
Diabetes		1 (0.1)	1 (0.2)
Prevalence of iliac aneurysms in the Korean cohort			
		Men, n = 1343	Women, n = 1747
Iliac aneurysm		12 (0.9)	42 (2.4)
Age	≤65 years	5 (0.4)	16 (0.9)
	>65 years	7 (0.5)	26 (1.5)
Right		7 (0.5)	26 (1.5)
Left		7 (0.5)	24 (1.4)
Bilateral		2 (0.1)	9 (0.5)

**Table 3 jcm-12-00484-t003:** Underlying conditions in the Korean participants with abdominal aortic aneurysm.

			Unadjusted		Adjusted	
n = 22		Number (%)	OR (95% CI)	*p* *	Adjusted OR (95% CI)	*p*
Sex	Women †	1 (4.5)	0.036 (0.051–0.268)	0.001	0.158 (0.015–1.685)	0.126
	Men	21 (95.5)				
Age †		74.5 (70.75–80.25 **)	1.102 (1.044–1.163)	<0.001	1.096 (1.037–1.159)	0.001
BMI		24.5 (21.17–26.74 **)		0.875		
Hypertension		4 (18.2)		0.853		
Diabetes		13 (59.1)		0.291		
Cerebrovascular disease		5 (22.7)	2.907 (1.065–7.938)	0.037		
Chronic kidney disease †		3 (13.6)	7.152 (2.067–24.752)	0.002	6.135 (1.694–22.227)	0.006
Smoking history †		20 (90.9)	19.018 (4.437–81.517)	<0.001	5.874 (1.054–32.733)	0.043

* *p*, logistic regression analysis. ** interquartile range. † independent variables used in the adjusted model. OR, odds ratio; CI, confidence interval.

**Table 4 jcm-12-00484-t004:** Linear and multilinear regression of infrarenal aortic diameter with variables in the Korean participants.

	Linear Regression		Multilinear Regression	
Variables	Coefficient (95% CI)	*p* *	Coefficient (95% CI)	*p*
Female sex	−1.640 (−1.841, −1.438)	<0.001	−0.724 (−1.032, −0.417)	<0.001
Age	0.010 (−0.001, 0.021)	0.001		0.058
BSA *	5.459 (4.781, 6.137)	<0.001	3.444 (2.566, 4.322)	<0.001
Hypertension		0.889		
Diabetes	0.435 (0.219, 0.651)	<0.001	0.361 (0.148, 0.573)	0.001
Hyperlipidemia	0.406 (0.195, 0.617)	<0.001		0.073
Coronary artery disease	0.361 (0.120, 0.602)	0.003	0.409 (0.165, 0.654)	0.001
Smoking history	1.429 (1.212, 1.645)	<0.001		0.051
Alcohol	1.263 (1.048, 1.478)	<0.001	0.424 (0.153, 0.696)	0.002

* Because of the multicollinearity, only the body surface area was used for multilinear regression analysis instead of the body mass index. BSA, body surface area.

## Data Availability

The data presented in this study are available on request from the corresponding author. The data are not publicly available due to ethical restrictions.

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
