# Peer review of "Prevalence and Risk Factors of Abdominal Aortic Aneurysms Detected with Ultrasound in Korea and Belgium"

_jcm, 2023, doi:10.3390/jcm12020484_

Round 1

Reviewer 1 Report

I congratulate with the authors for their efforts in conducting this original comparison between the Korean and the Belgian AAA population.

Following are my few suggestions to improve the manuscript.

Abstract: you can spare some words (for instance, you repeat twice 3124 participants), while you should include pulmonary disease and body mass index, which in some way are correlated to the pathogenesis of AAA.

Table 1: please, report the meanings of the acronyms at the bottom.

Furthermore, you should comment all the statistically significant p values in the Results section.

Discussion, lines 323-326. I would use caution when assessing some correlation between the risk of rupture and a AAA diameter of 30 mm.

Author Response

Thank you for your comment. We have revised our manuscript according to your suggestions.

I congratulate with the authors for their efforts in conducting this original comparison between the Korean and the Belgian AAA population.

Following are my few suggestions to improve the manuscript.

Abstract: you can spare some words (for instance, you repeat twice 3124 participants), while you should include pulmonary disease and body mass index, which in some way are correlated to the pathogenesis of AAA.

Thank you for your comment. We have deleted one of the duplicates ‘3124’. We included pulmonary disease and body mass index.

Table 1: please, report the meanings of the acronyms at the bottom.

Furthermore, you should comment all the statistically significant p values in the Results section.

 Thank you for the comment. We have added the meanings of the acronyms of the Table 1. We commented all the statistically significant p values in the Results section.

Discussion, lines 323-326. I would use caution when assessing some correlation between the risk of rupture and a AAA diameter of 30 mm.

 Thank you for the important comment. We agree with the reviewer’s point that the sentence may be misleading. In order to clarify the meaning that AAA's diameter of 30mm corresponds to ASI 1.5, the sentence has been modified as follows; “the suggested cut off to define AAA would be an ASI of 1.5 corresponding to 30 mm AAA in men.”

Reviewer 2 Report

Overall, I am not sure about the significance of this study. The only important thing is that perhaps in Asian/Pacific Island populations, aortic index rather than aortic size should be used to detect/define AAA. 

Author Response

Thank you for your comment. Considering that AAA is associated with significant morbidity and mortality due to rupture, it seems uncontroversial that AAA is an important disease to screen at the national level, especially in selected subgroups.

Overall, I am not sure about the significance of this study. The only important thing is that perhaps in Asian/Pacific Island populations, aortic index rather than aortic size should be used to detect/define AAA. 

We believe that these findings contribute with meaningful results, among some; the ASI differences as well as the differences in aortic diameters. Very little has been reported from Asian populations regarding aortic diameters, prevalence of AAA or CIA, especially including also women. This paper is an attempt to clarify and report some basic findings from a large population in this context, which also includes both women and men in a modern setting. Screening for AAA has been shown to save lives, and to be cost effective to perform in older men, with a prevalence above 0.5%. This could be true also in an Asian country, if the prevalence is high enough, and the program can be performed at a reasonable cost. This study does contribute with some baseline data for such an evaluation.

Reviewer 3 Report

1. In your demographic comparison data, there are significantly more ex-smokers in the Korea population versus the Belgium population. Is there another tracked population that has an ex-smoker population that is more similar than the Belgian cohort? Smoking has been shown to be related to aneurysm size and severity as well as several of the other demographic factors can also be affected by smoking. 

2. Based on figure 4, it appears that the men in both the Korean and Belgian populations start out with roughly similar aortic size but then age tends to affect the size of the Belgian aorta much more than the Korean. Maybe a mention of these differences should be considered as part of the discussion. 

Author Response

  1. In your demographic comparison data, there are significantly more ex-smokers in the Korea population versus the Belgium population. Is there another tracked population that has an ex-smoker population that is more similar than the Belgian cohort? Smoking has been shown to be related to aneurysm size and severity as well as several of the other demographic factors can also be affected by smoking. 

Thank you for the comment. The high proportion of “ever-smokers” in Asia is very unfortunate. Among the men with AAA the absolute majority were “ever smokers”, 19/22, 86%.

It would be interesting to examine larger cohorts of never smokers in Korea, in order to investigate if the already quite low prevalence would be even lower. Since this is a highly relevant question, we will conduct subsequent studies in order to better consider such influential factors in AAA cohorts, especially in Asia.

  1. Based on figure 4, it appears that the men in both the Korean and Belgian populations start out with roughly similar aortic size but then age tends to affect the size of the Belgian aorta much more than the Korean. Maybe a mention of these differences should be considered as part of the discussion. 

Thank you for the important comment. We have added this finding in the Discussion section as the reviewer suggested as follows; “Interestingly, the difference in the infrarenal aortic diameter in the age group of 51-55 year was similar between the Korean and Belgian Groups, as compared to the detected differences in the older age groups (Figure 4-a). Age seemed to affect the aortic diameter of the Belgian cohort more than the Korean cohort.”
